# Adenine-Induced Nephropathy Reduces Atherosclerosis in *ApoE* Knockout Mice

**DOI:** 10.3390/biom12081147

**Published:** 2022-08-19

**Authors:** Laeticia Scherler, Sofia N. Verouti, Daniel Ackermann, Bruno Vogt, Geneviève Escher

**Affiliations:** 1Department for BioMedical Research (DBMR), University of Bern, 3010 Bern, Switzerland; 2Department of Nephrology and Hypertension, Inselspital, Bern University Hospital, University of Bern, 3010 Bern, Switzerland

**Keywords:** chronic kidney disease, cholesterol efflux, reverse cholesterol transport, atherosclerosis, tibia

## Abstract

Background: Cardiovascular events are the main cause of death in patients with chronic kidney disease. We hypothesize that the protective effects of renal cholesterol and vitamin D_3_ metabolism are lost under this condition. Nephropathy was induced by adenine in Apolipoprotein E knockout mice. The atherosclerotic phenotype was compared to mice with normal renal function. Methods: Mice were fed a western diet ±0.15% adenine. Urine and feces were collected to assess renal function and fecal output. Atherosclerosis, serum lipoprotein composition and functionality, hepatic lipids, and expression of genes involved in lipid metabolism, vitamin D_3_ and Na^+^ homeostasis, were assessed. Bones were analyzed by microCT. Results: Mice fed with adenine showed enhanced urinary Na^+^, Ca^2+^, and Pi excretion, reduced urinary pH, Urea_Urine_/Urea_Serum_, and Creatinine_Urine_/Creatinine_Serum_ ratios. They developed less atherosclerosis. Lipoproteins in serum and hepatic lipids remained unchanged. Cholesterol efflux increased. Fecal output of cholesteryl ester and triglycerides increased. In the liver, mRNA levels of *Cyp27a1*, *Cyp7a1*, and *Scarb1* increased; in the kidneys, *Slc9a3*, *Slc12a3, Vdr*, and *Cyp24a1* decreased. Adenine increased cholesterol efflux in vitro. Tibias were shorter. Conclusion: Adenine induced tubular damage and was athero-protective because of enhanced cholesterol efflux and lipids elimination in feces. Bone growth was also affected.

## 1. Introduction

Cardiovascular disease (CVD) is the prevalent cause of mortality and morbidity in western countries and is also dramatically increasing in low and middle-income countries [1]. The main condition underlying CVD is atherosclerosis [2]. This is more than a lipid storage disease; it is a low-grade inflammation of the vascular wall. The endothelial cells lining blood vessels become chronically activated through the combination of turbulent blood flow, lipid accumulation, and exposure to inflammatory mediators such as IL-1β [3]. The deposition of lipids combined with the accumulation of T-cells and macrophages occurs over decades and leads to an endothelial injury response [4]. During this process, reactive oxygen species account for the oxidation of low-density lipoprotein (LDL) and polyunsaturated fatty acids, which in turn accumulate in the vascular wall. This causes damage to cellular components and promotes inflammation by activating pro-atherogenic transcriptional factors such as PU1, CEBPB, and AP-1. Once atherosclerotic plaques are formed, they might rupture. This initiates a thrombotic cascade leading to the damage of downstream vessels. Atherosclerosis is accelerated in patients with metabolic syndrome, a clustering of at least three of the following medical conditions: visceral obesity, hypertension, hyperlipidemia consisting of high serum triglycerides and low HDL, hyperglycemia, and hyperinsulinemia with insulin resistance [5]. The worldwide increase in the prevalence of this syndrome is attributed to profound changes in lifestyle, including low physical activity, excessive smoking combined with a high fat regimen containing large amounts of salt and sugar, and low levels of potassium and vitamin D [6].

During atherosclerosis formation, cholesterol accumulates in the vessel wall because of an imbalance between pathways delivering cholesterol to vascular cells and those removing it. Pharmacological means have been developed to regulate the delivery of cholesterol to cells. Although they are widely applied for the treatment of atherosclerosis, cardiovascular events remain unacceptably high worldwide [7]. To achieve further therapeutic progress, targeting specific steps of reverse cholesterol transport (RCT) would potentially be of benefit. RCT is responsible for the removal of cholesterol from peripheral tissues followed by its transfer via plasma to the liver for either recycling or excretion from the body as part of bile acids [8]. The first and most likely rate-limiting step of RCT is cholesterol efflux, the transfer of cholesterol from cells to an extracellular acceptor, specifically to high density lipoprotein (HDL).

Apolipoprotein E (*ApoE*) is a component of all lipoproteins except LDL and interacts with LDL receptors and heparan sulphate proteoglycans (HSPG) in the liver, and is thus essential for the clearance of triglyceride-rich lipoproteins from the circulation [9]. The genetic deficiency of *ApoE* leads to the accumulation of cholesterol-rich remnants in plasma in both humans and rodents. *ApoE* deficient mice show severe hypercholesterolemia and develop atherosclerosis in a short time. The process is accelerated by western diet (WD) feeding [10]. Addition of cholic acid (CA) worsens atherosclerosis by changing lipid composition: it increases LDL decreases HDL, and enhances intestinal cholesterol absorption [11,12].

Patients with chronic kidney disease (CKD), another growing health burden worldwide, are at high risk of CVD morbidity and mortality, even at young age [13]. However, CVD is mainly the leading cause of death in dialyzed patients and not renal failure [14]. The worldwide increased prevalence of CKD is linked to the emergence of metabolic syndrome. Signs of kidney dysfunction are increased levels of creatinine, urea, K^+^, and fibroblast growth factor (FGF-23) in serum [15]. Renal dysregulation is also known to disturb mineral metabolism [16]. CKD patients tend to have higher plasma triglycerides and lower HDL [17]. Consequently, the athero-protective role of HDL in RCT is lost.

In mice, there are many strategies to induce CKD. The most commonly used model consists of a reduction of renal mass by 5/6 nephrectomy. This surgical intervention leads to a quite high mortality rate, mostly because of ureteral injury and bleeding [18]. In addition, the experimental variance of the model due to differences in the size of the remnant kidney for the 5/6 nephrectomy significantly impacts the experimental outcome. In contrary, the adenine-induced nephropathy can be obtained by simply supplementing adenine directly in the diet [19]. Adenine is a purine base normally converted into the uric acid allantoin, which is excreted in urine. In excess, adenine accumulates and is converted into 8-dihydroxy adenine and eventually 2,8-dihydroxyadenine. These non-soluble materials are crystallized in renal tubules and cause renal damage [20]. Adenine-induced nephropathy has a high survival rate due to the non-invasive method. Mice show signs of urea changes in plasma, interstitial fibrosis, cysts in tubules, mineral and bone disorder, secondary hyperparathyroidism, and increased levels of FGF-23 [19,21]. The latter regulates phosphate homeostasis and Vitamin D metabolism, thus increasing to compensate for the pathophysiological changes [22].

In the present study, we analyzed the contribution of a functional kidney to the protection against atherosclerosis development. For this aim, we took advantage of the adenine-induced renal failure and studied the atherosclerotic phenotype and bone phenotype in *ApoE* KO mice fed with a pro-atherogenic diet WD containing 0.025% CA (WD-CA) with normal or impaired renal function.

## 2. Materials and Methods

### 2.1. Materials

Cholic acid (CA) (C1129), Oil red O (O0625), hematoxylin solution Gill no. 3 (GHS316), and eosin Y solution aqueous (HT110216) were from Sigma-Aldrich (St. Louis, MO, USA).

### 2.2. Mice

Animal experimentation was approved by the Ethics Committee for Animal Experiments of the Veterinary Service of the Canton of Berne (BE61/17 and BE58/19) and conformed to the rules of the Swiss Federal Act on Animal Protection.

*ApoE* KO mice (Charles River Laboratories, Sulzfeld, Germany) on a C57BL/6J background were bred and weaned according to the rules at the central animal facility of the University of Bern.

Mice were maintained under 12-h dark–light cycles with unrestricted access to food and water. Only males were selected for the study due to gender-specific differences in response to CKD [23]. At the age of 5 weeks, males were fed for 2 weeks with a WD containing 21% fat and 0.15% cholesterol (D-12079B, Kliba-nafag, Granovit AG, Switzerland) supplemented with 0.025% CA and 0.1% cacao (WD-CAC). Cacao was used to mask the taste of adenine. Mice were randomized into 2 groups. One group was fed for the following 5 weeks with the same diet (WD-CAC), and the other was fed with WD-CAC supplemented with 0.15% adenine to induce nephropathy.

Five days before sacrifice, mice were placed in metabolic cages. After 3 days of acclimatization, urine and feces were collected over 24 h for 2 consecutive days. Urine was collected under mineral oil. Mice were starved 4 h before sacrifice under isoflurane inhalation for blood and organ collection (kidney, liver, heart, aorta, tibia, and femur). Blood was centrifuged at 4 °C for 15 min at 13,000 rpm and serum stored at −20 °C until analyzed. Tissues were either frozen in liquid nitrogen and stored at −80 °C or fixed in 4% paraformaldehyde in phosphate-buffered saline (PFA) for 24 h and transferred to PBS and stored at 4 °C.

### 2.3. Urine, Serum, and Tissue Biochemistry

Urinary electrolytes were determined at the core laboratory of the Bern University Hospital, Bern, Switzerland.

Serum total cholesterol (Fujifilm Wako Pure Chemical Corporation, Hongkong, China and Sigma-Aldrich, Missouri, MI, USA), low-density lipoprotein (LDL) and high-density lipoprotein (HDL) were quantified by the HDL and LDL/VLDL cholesterol assay kit (Abcam, Cambridge, UK), triglycerides with a kit from Wako Chemicals GmbH (Neuss, Germany), and cholesterol and cholesteryl esters (CE) with a CE quantification kit from Calbiochem-Merck (Millipore, Zug, Switzerland). The QuantiChromTM assay kit from BioAssay Systems (Hayward, CA, USA) was used for the measurement of creatinine, and the FGF-23 ELISA kit (Abcam) to quantify FGF-23.

In liver and feces homogenates, TG was assayed using a TG quantification kit (BioVision, Mountain View, CA, USA), and cholesterol and cholesterol esters with the same quantification kit as for serum.

### 2.4. Histology

Kidneys were analyzed in paraffin sections of 5 μm and stained with hematoxylin and eosin (H&E). Slides were scanned with the Panoramic 250 Flash II (3DHistec, Budapest, Hungary).

Heart sections were prepared for frozen sections in O.C.T compounds (Tissue-Tek^®^ Sakura, Netherlands). Sections of 5μm were cut from the aortic root with a Cryostat C60 Hyrax (Carl Zeiss, Germany) and stained with Oil Red O (ORO) and hematoxylin. Slides with lesions in the aortic valves were scanned as described above and quantified on QuPath (V0.2.0) (University of Edinburgh, Edinburgh, Scotland) in a blinded manner. The area ratio between the diameter of the aortic root and atherosclerotic lesions was measured for each section, and the mean was calculated.

### 2.5. RNA Extraction and Real-Time PCR in Kidney and Liver

RNA was extracted with TRIzol Reagent (Invitrogen Reagent, Massachusetts, MA, USA) using 100 mg of liver or kidney. Reverse transcription was performed with 2 µg of RNA using the PrimeScriptTM RT Reagent kit from Takara (Bio Inc., Kusatsu, Japan) and hexanucleotide mix from Roche Diagnostics (Basel, Switzerland). For quantitative real-time PCR, primers and probes were obtained from Microsynth (Balgach, Switzerland), and, respectively, from Roche Diagnostics (Basel, Switzerland), and assays on demand from Thermo Fisher (Waltham, MA, USA). The following primers were used: *Vdr*: (5′-CACCTGGCTGATCTTGTCAGT-3′ and R: 5′-CTGGTCATCAGAGGTGAGGTC-3′, probe 89), *Slc9a3*: F: 5′-TCCATGAGCTGAATTTGAAGG-3′ and R: 5′-TACTTGGGGAGCGAATGAAG-3′ probe 5, *Aqp2*: NM_009699.3, *Slc12a3*: NM_001205311.1, cyp2r1: NM_177382.3), *Cyp27a1*: Mm00470435_g1, *Cyp27b1*: Mm01165918_g1, *Cyp24a1*: Mm00487244_m1, *Ldlr*: Mm01177349_m1, *Cyp7a1*: Mm00484152_m1, *Cyp3a11*: Mm00731567_m1, *Cyp8b1*: Mm00501637_s1, *Cav1*: Mm01129316_m1, *Srb1*: Mm00450234_m1, *Abca-1:* Mm00442663_m1) as FAM dyes. Β-actin was used as a housekeeping gene (NM_007393.1 VIC Thermo Fischer). The reaction was performed with GoTaq^®^ Probe qPCR Master Mix (Promega Corporation, Madison, WI, USA) with 100 ng/cDNA/reaction on a QuantStudio 1 RT-PCR system. Data was processed with the QuantStudioTM Design & Analysis software (v.1.5.0) (Thermo Fisher Scientific). Quantification was performed by the relative quantification method using *ApoE* KO as a calibrator.

### 2.6. MicroCT Scan

Bones were fixed in 4% paraformaldehyde in phosphate-buffered saline for 24 h, washed for 4 h in distilled water and stored in 70% ethanol for MicroCT scan (Scanco Medical AG, Brüttisellen, Switzerland) with an X-ray source set at 70 kVp and 55 μA at high resolution. The diameter of the sample holder was 12.3 mm, which allowed a resolution of 6 μm. The integration time was set at 300 ms. The task chosen for 3D evaluation of 2D slides was bone trabecular morphometry (script from Scanco Medical AG, Brüttisellen, Switzerland) with Gauss sigma at 0.8, Gauss support at 1, and low/high threshold at 220/1000.

### 2.7. Cholesterol Efflux

Cholesterol efflux was conducted as described previously [24], using mouse serum as an acceptor. RAW 264.7 (ATCC, USA) cells were grown in 12-well plates at a concentration of 10^5^ cells/well and cultured in RPMI 1640 Gibco^®^ (Thermo Fisher) containing L-glutamine, supplemented by 10% fetal bovine serum (FBS), 100 μg/mL penicillin/streptomycin, and 2 mM glutamine. Cells were labelled for 48 h with 1 mCi/mL [1α,2α-^3^H]-cholesterol (Anawa, Kloten, Switzerland), washed 3 times with phosphate buffered saline (PBS) (GibcoTM), and equilibrated overnight with OPTIMEM (Thermo Fisher). For cholesterol efflux, cells were incubated for 2 h with serum-free RPMI containing 2% mouse serum. The medium was collected and centrifuged for 15 min at 4 °C at 2500 rpm. Cells were harvested in 1 mL of distilled water. Radioactivity was counted in 100 µL aliquots in the medium and cells, and cholesterol efflux was calculated as the percentage of labeled cholesterol released to the medium divided by the amount of total cholesterol in the medium and cells in each well.

To test the effect of adenine on cholesterol efflux in vitro, dose-dependent concentrations of adenine were incubated during each phase of the assay (labelling, incubation in OPTIMEM, and the efflux). Cholesterol efflux was performed with a 2.5% plasma pool as an acceptor.

### 2.8. Statistical Analysis

Data were analyzed with GraphPad Prism 9.4 (GraphPad Software, Inc., San Diego, CA, USA). To determine statistically significant differences, unpaired t-test, or one-way ANOVA were used.

## 3. Results

### 3.1. Development of a New Model of Adenine-Induced Nephropathy with Mild Effect on Body Weight

To study the effect of CKD on atherosclerotic plaque formation, *ApoE* KO mice were fed for 2 weeks with a pro-atherogenic WD-CAC followed by 5 weeks with WD-CAC or WD-CAC with 0.15% adenine. At the end of the experiment, mice were placed in metabolic cages to monitor food and water intake and collect urine and feces. The same amount of food was consumed daily (Figure 1A), and the treatment had no effect on body weight (Figure 1B). At the time of sacrifice, kidney weight tends to be lower (Figure 1C), but liver weight was similar between the two groups (Figure 1D). Adenine-fed mice in metabolic cages drank twice as much (*p* < 0001) (Figure 1E), excreted five times more urine (*p* < 0.0001) (Figure 1F), and were two times less able to concentrate urine (*p* < 0.01) (Figure 1G), as indicated by the fluid balance. The urinary pH was more acidic in the adenine group (*p* < 0.001) (Figure 1F).

To estimate the impact of adenine on renal function, biochemical parameters were measured in urine and serum (Table 1). In urine, the 24 h net excretion of Na^+^, Ca^2+^, Mg^2+^, and Pi was significantly increased, whereas creatinine, K^+^, Cl^−^, urea, and glucose remained unchanged. Creatinine and blood urea nitrogen (BUN) were also constant in the serum, whereas osmolality, FGF23, and glucose were markedly increased. The renal function of both groups was then assessed using the Urea_Urine_/Urea_Serum_ and Creatinine_Urine_/Creatinine_Serum_ ratios; both ratios were significantly reduced (*p* < 0.001) in mice fed with adenine, indicating CKD induction (Table 1).

Morphological changes of the kidney were confirmed by histology in formalin-fixed kidney sections stained with H&E. *ApoE* KO mice fed with WD-CAC had normal renal histology (Figure 2A,B). In histological sections of adenine-treated mice, the appearance of glomeruli was normal (Figure 2C,D). However, we found intraluminal adenine crystals in the tubular compartment (see black arrow in Figure 2C) in ~70% of the sections analyzed. In the presence of adenine crystal deposition, tubular epithelial cells showed signs of tubular damage with apoptotic cell nuclei (dashed arrow). These signs of tubular damage were also present in the absence of adenine crystal deposition (Figure 2D), as indicated by dilated tubules (small, dotted arrows), cell debris (fat black arrow), and tubular vacuolization (dashed arrow). Taken together with the biochemical results obtained from metabolic cages, adenine-induced nephropathy was achieved by tubular damage that does not affect animal growth.

### 3.2. The Anti-Atherosclerotic Effect of Adenine Is Mediated by Enhanced Cholesterol Efflux

The consequence of the reduced renal function induced by adenine on atherosclerosis development was evaluated by quantifying atherosclerotic plaque in aortic valve sections of *ApoE* KO mice 5 weeks after starting with adenine (Figure 3). Lesions were quantified in serial sections (Figure 3A,B), and the percentage of lesions was calculated against the total vessel area (Figure 3C). There was ~60% (*p* < 0.001) less atherosclerosis in mice fed with adenine.

To pinpoint the protective atherosclerotic mechanism in the adenine group, the serum lipoprotein profile was analyzed (Table 2). Total cholesterol, HDL, and LDL, as well as the HDL/LDL ratio and triglycerides remained unchanged in both groups. To test the effect of adenine-induced nephropathy on HDL functionality, cholesterol efflux was performed in RAW264.7 cells using serum from experimental animals. Cholesterol efflux increased by 40% (*p* < 0.0001) when serum from mice fed with adenine was used (Figure 4A). Interestingly, adenine itself at a high concentration increased cholesterol efflux (Figure 4B).

### 3.3. Effect of Adenine on Hepatic Lipid Composition and Cholesterol Excretion

The content of hepatic cholesterol, cholesteryl ester, and triglycerides of *ApoE* KO mice with adenine-induced nephropathy remained unchanged when compared to those with normal renal function (Table 2). In contrast, changes were observed in fecal output (Table 2). Whereas total cholesterol excretion was not significantly increased, elimination of cholesteryl esters and triglycerides was enhanced twice (*p* < 0.01), and four times (*p* < 0.001), respectively, in mice with tubule disfunction.

### 3.4. Impact of Adenine-Induced Nephropathy on Gene Expression in Kidney and Liver Tissues

First, we confirmed the induction of adenine-induced nephropathy by analyzing the mRNA levels of genes involved in the maintenance of water balance and pH in the kidney (Figure 5A). The mRNA for *Slc9a3*, the gene encoding the sodium-hydrogen antiporter 3 (NHE3) in the proximal tubule, was reduced by more than 50% (*p* < 0.01) in *ApoE* KO mice fed with adenine. The mRNA level of the sodium-chloride cotransporter (NCC) (*Scl12a3*) located in the distal convoluted tubule was also down-regulated twice (*p* < 0.01). Finally, the expression of *Aqp2*, the gene encoding for Aquaporin 2 in the principal cells of the connecting tubule and collecting duct, was slightly reduced but not statistically significant.

Since vitamin D deficiency is a hallmark of CKD, we quantified the expression of the vitamin D receptor (*Vdr*), the ligand for the active vitamin D metabolite 1,25-dihydroxyvitamin D_3_ (1,25(OH)_2_D_3_), as well as the two genes involved in the metabolism of vitamin D in the kidney, *Cyp27b1*, which catalyzes the conversion 25(OH)D_3_ to 1,25(OH)_2_ D_3_, and *Cyp24a1*, responsible for the inactivation of 1,25(OH)_2_D_3_ into 24,25(OH)_2_D_3_ (Figure 5B). There was a 40% and 60% reduction in the mRNA level of *Vdr* (*p* < 0.05) and *Cyp24a1* (*p* < 0.01) in *ApoE* KO mice with adenine-induced nephropathy. For *Cyp27b1*, the decrease was not significant.

Next, we investigated in liver tissues the expression of the genes involved in cholesterol homeostasis (Figure 5C). The mRNA levels of the cytochromes *Cyp27a1* and *Cyp7a1* involved in the first step of the alternative and neutral pathway of bile acid synthesis were upregulated by 40% (*p* < 0.05) and 65% (*p* < 0.01) respectively, in the adenine group. The expression of *Cyp8b1*, a key determinant in the production of the bile acid CA, and *Cyp3a11*, involved in bile acid detoxification, was not changed. The expression level of the gene encoding for the LDL receptor (*ldlr*) was not changed either. When the genes involved in cholesterol efflux were quantified, *Scarb1* encoding for the scavenger receptor class B type 1 (SR-B1) slightly increased (*p* < 0.01). The *Cav-1* encoding for Caveolin 1 and *Abca1* for the ATP-binding cassette A1 (ABCA1) did not significantly increase. Finally, we quantified the relative expression of *Cyp2r1*, working in tandem with *Cyp27a1*, which hydroxylates vitamin D_3_ into 25(OH)D_3_ in the liver (Figure 5D). The mRNA level of *Cyp2r1* did not increase significantly but there was a 40% increase of *Cyp27a1* (*p* < 0.05), the cytochrome accounting for the first step of bile acid synthesis.

Last, we performed gene analyses in the colon. The expression level of *Cy27a1*, *Scarb1*, *Cav*, and *Abca1* was unchanged (Figure 5E). This was also the case for the two sterol transporters, *Abcg5* and *Abcg8.* The same was true for *Vdr* and *Cyp27b1* (Figure 5F).

### 3.5. Effect of Adenine on Bone Structure

Since bone mineralization is disturbed in patients with CKD, we evaluated the impact of adenine on bone structure in *ApoE* KO mice in our experimental conditions (Figure 6). We first measured the length of tibias and femurs and found that tibias were significantly smaller (*p* < 0.05) in mice with nephropathy (Figure 6A), whereas the length of femur remained intact (Figure 6B). We next analyzed by µCT scan the distal part of the femurs to determine the bone volume (BV) (Figure 6C), the trabecular thickness (Tb. Th) (Figure 6D), the trabecular number (Th. *n*) (Figure 6E), and the trabecular space (Th. sp) (Figure 6F). Changes were scarce, with only a tendency to a decline in bone density, indicated by a slight decrease in the BV/TV percentage (Figure 6C) and an increase in trabecular space (Figure 6F). With 3D imaging, no significant changes in morphology were monitored (Figure 6G,H).

## 4. Discussion

The mortality rate in patients suffering from reduced glomerular filtration rate is associated with cardiovascular risk, and for those with CKD, death is more likely to occur than end-stage renal disease [25]. Progressive kidney disfunction also accounts for the disruption of bone and mineral metabolisms [16]. To investigate the anti-atherosclerotic role played by kidneys and their contribution to the preservation of bone structure, we took advantage of the atherosclerotic murine model, the *ApoE* KO mice fed with a pro-atherogenic diet consisting of WD supplemented with CA. We induced a decline in renal function with adenine and analyzed their atherosclerotic and bone phenotypes.

The key findings of our study are summarized in Figure 7. *ApoE* KO mice fed with adenine showed a reduced renal function characteristic for a tubular dysfunction. They were unexpectedly protected against the development of atherosclerosis, mostly because of an increase in cholesterol efflux and enhanced elimination of cholesterol metabolites in feces. Adenine itself increased cholesterol efflux in vitro. Tibia growth was slowed down because of increased FGF23 in serum and Ca^2+^ and Pi loss in urine.

To our knowledge, this is the first study where adenine nephropathy is induced in combination with WD-CA to enhance the atherogenicity of the diet. Moreover, the daily ingestion of adenine within the food was enough to trigger the induction of CKD. As described by Jia et al. [19], we started in a previous study with an induction phase consisting of 10 days with 0.3% adenine, followed by a reduction to 0.2% and 0.15% as a maintenance phase. Under these conditions, the induction phase led to severe weight loss. We then reduced the dose of adenine to 0.15% and masked its taste by adding 0.1% of cocoa in WD. In the end, we were able to obtain similar food intake and body weight between the two groups and could conduct our long-term experiment on atherosclerosis.

The decline in renal function was confirmed at many levels. First, renal function was calculated using the Urea_Urine_/Urea_Serum_ and Creatinine_Urine_/Creatinine_Serum_ ratios. Both ratios were significantly reduced, indicating a reduced renal clearance rate in mice fed with adenine (Table 1). Urea appears to be a more accurate marker of uremia than creatinine, whereas the ratio of creatinine/body weight followed the urea levels [19,26]. Unlike in humans, creatinine in mice is secreted by the tubules rather than filtrated by the glomeruli [27]. Second, *ApoE* KO mice fed with adenine had elevated FGF23 in serum (Table 1). FGF23 is involved in the regulation of Pi and 1,25(OH)_2_D_3_, and its upregulation points to a compensatory mechanism to overcome the Pi accumulation and maintain the balance of bone metabolism and catabolism [22]. Third, tubular damage induced by adenine was demonstrated by histological changes consisting of tubular damage and crystal deposition of adenine (Figure 2). Fourth, gene expression analysis in renal tissues revealed a down-regulation of genes involved in Na^+^ reabsorption, mainly NHE3 and NCC (Figure 5A), indicating sodium and proton loss in the urine. The low pH in urine is also an independent predictor of CKD [28]. Fifth, in metabolic cages, *ApoE* KO mice fed with adenine drank significantly more water and excreted more urine but were unable to maintain their fluid balance due to concentrating tubular defects (Figure 1F–H). Furthermore, the size of their kidneys tended to shrink (Figure 1C). Finally, osmolality in serum was reduced due to tubular damage (Table 2). This salt and water-losing phenotype is in line, although with less severity, with those observed by Dos Santos et al., in rats [29]. The renal failure obtained in our study is not as severe as the one found in patients with CKD. The most common cause of CKD in humans is glomerular damage, which is not the case in adenine-induced nephropathy. Therefore, this model should not be regarded as a model of CKD, per se, but rather as a complementary model of renal failure.

Quantification of atherosclerotic lesions in the aortic root surprisingly identified *ApoE* KO mice with adenine-induced nephropathy as being protected (Figure 3). This finding is mainly attributed to (i) elevated cholesterol efflux (Figure 4) and (ii) increased fecal output of cholesteryl ester and triglycerides (Table 2). Indeed, the amount and composition of lipoprotein in serum, which is a good predictor of atherosclerotic risk, was not significantly affected by the adenine treatment. However, the ability of mouse serum to remove cholesterol from cells in our cholesterol efflux system increased by 40%. Adenine itself enhanced cholesterol efflux in vitro (Figure 4). Adenine is a purine nucleobase involved in many biochemical functions and also a precursor of the second messenger cAMP, a well-characterized stimulator of cholesterol efflux [30]. Nevertheless, the direct link between adenine and cholesterol efflux was never shown. In liver tissues, the mRNA levels of *Cyp27a1* and *Scarb1*, two genes of the cholesterol efflux pathway, were significantly increased, whereas *Cav-1* and *Abca1* were not significantly changed (Figure 5C). Cholesterol efflux is the first and rate-limiting step of RCT, and dysfunction of genes involved in these pathways are linked to atherosclerosis [31]. As a corollary, upregulation of these genes provides athero-protection. The SR-B1 acts as a multifunctional receptor for cholesterol influx and efflux [32]. In the liver, it mediates the selective uptake of HDL-derived cholesteryl esters into the cells. The enhanced cholesterol clearance in the circulation in mice with adenine-induced nephropathy can also be explained by the increase of *Cyp7a1*. *Cyp7a1,* which converts cholesterol to 7-hydroxycholesterol, the rate-limiting step in bile acid biosynthesis, promotes the production of bile acids, which are key components of RCT. The lipid content in the liver was not affected by adenine (Table 2). An analysis of fecal excretion collected from mice placed in metabolic cages points to an additional athero-protective effect of adenine in the intestine, with cholesteryl esters and triglycerides being more excreted. A previous study in rats already pointed to a mild effect of adenine on the gastrointestinal tract and its function [33].

In our model of adenine-induced nephropathy, only mild changes were observed in bone structure (Figure 6). Although longitudinal growth of tibias was decreased (Figure 6A), bone volume (Figure 6C) and trabecular thickness and number (Figure 6D,E) measured in femurs tended to be smaller, whereas the trabecular space tended to be increased (Figure 6F). Nevertheless, these moderate changes can be attributed to the elevation of circulating FGF23, an important determinant of Pi homeostasis, vitamin D metabolism, and bone mineralization. Studies in FGF23 transgenic animals showed that tibial bone length was markedly reduced and its mineralization impaired [34]. Similarly, *Vdr* KO mice develop typical features of rickets [35], whereas *Cyp27b1* KO mice suffer from severe growth retardation, hypocalcemia, and poor bone mineralization [36]. In our *ApoE* KO mice with adenine-induced nephropathy, FGF23 was increased, and *Vdr* and *Cyp27b1* mRNA levels were down-regulated in the kidney, providing evidence for impaired bone growth conditions. Furthermore, to compensate for the decrease in vitamin D metabolism, *Cyp24a1* was downregulated in the kidney to avoid the conversion of 1,25(OH)_2_D_3_ into 24,25(OH)_2_D_3_.

As a potential limitation, all experiments were performed with male mice, so potential sex differences remain to be determined. Furthermore, analyzing mice at the age of 12 weeks was probably too early to detect strong changes in the bone phenotype. Since aging and estrogen deficiency play a pivotal role in degenerative bone disease, a similar study should be conducted after bilateral ovariectomy in elderly mice. Because of the limitation in sample size in serum, we could not measure PTH and the different metabolites of vitamin D such as 25(OH)D_3_, 1,25(OH)_2_D, and 24,25(OH)_2_D_3_. Our adenine-induced nephropathy was rather mild and did not mirror the complete loss of renal function as seen in patients with CKD.

Further experiments to explore vitamin D metabolism in these mice should be conducted, such as supplementing WD with vitamin D_3_ or 25(OH)D_3_. Regarding the protective role of these metabolites against atherosclerosis and on bone mineralization, a similar procedure should be applied in *ApoE* KO mice with reduced kidney function. Investigations in females without or with bilateral ovariectomy to enhance unbalanced bone metabolism would complete the study. Lastly, since adenine itself increases cholesterol efflux, the first and rate-limiting step of RCT, adenine-induced nephropathy might not be the ideal model to assess the interaction between kidney disease and atherosclerosis. Whether adenine could be used in humans at a low dose to promote cholesterol efflux without causing kidney damage remains to be tested.

## 5. Conclusions

Our study shows that adenine-induced nephropathy could be achieved in *ApoE* KO mice fed with an atherogenic diet without severe weight loss. Mice suffered from polydipsia and polyuria with Na^+^, Ca^2^, and Pi loss. The tubular damage in kidneys was characterized by the presence of adenine crystal deposition and dilated tubules in histology sections. Under these conditions, mice were surprisingly less prone to developing atherosclerosis, mainly because their sera were more efficient to efflux cholesterol, and they excreted more lipids in their feces. In in vitro studies, adenine itself proved to be athero-protective because of its effect on cholesterol efflux, the first and rate-limiting step of RCT. These conditions were not as severe as those found in patients with CKD and hardly affected bone structure.

## Figures and Tables

**Figure 1 biomolecules-12-01147-f001:**
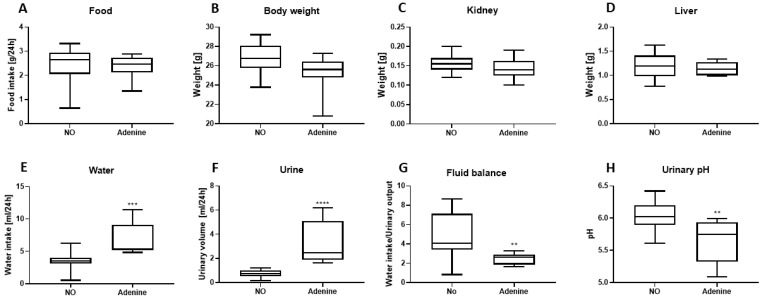
Mice follow-up in metabolic cages and at time of sacrifice. (**A**) Food intake; (**B**) Body weight; (**C**) Kidney weight; (**D**) Liver weight; (**E**) Water intake; (**F**) Urinary excretion; (**G**) Fluid balance calculated as water intake/urine volume, and (**H**) Urinary pH. NO represents the control group, fed with WD-CAC. Data are plotted as median with whiskers indicating Min and Max values (*n =* 9–11), with ** *p* < 0.01, *** *p* < 0.001 and **** *p* < 0.0001 for statistical difference.

**Figure 2 biomolecules-12-01147-f002:**
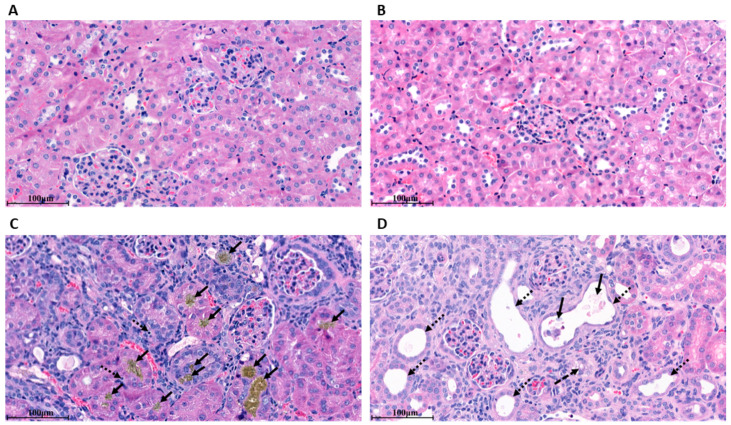
Representative light microscopy images of Hematoxylin-Eosin (H&E) staining of kidney tissues. *ApoE* KO mice were fed for 2 weeks with WD-CAC, followed by 5 weeks with WD-CAC ± adenine, and kidneys were harvested and processed for paraffin sections. (**A**,**B**) Two representative stainings of healthy kidney cortex sections with normal glomeruli and tubules. In adenine treated mice, the left panel (**C**) presents adenine crystal deposition (black arrow) and apoptotic cell nuclei (dotted black arrows), and the right panel (**D**), signs of acute tubular necrosis with dilated tubules (dotted black arrows), cell debris (fat black arrow), and tubular vacuolization (dashed black arrow). Scale bars *=* 100 μm.

**Figure 3 biomolecules-12-01147-f003:**
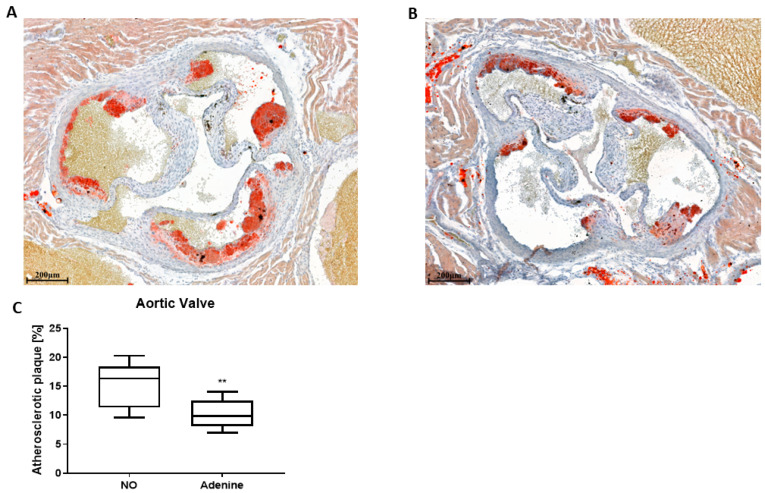
Adenine-induced nephropathy leads to a reduction of atherosclerotic lesions. *ApoE* KO mice were fed for 2 weeks with WD-CAC, followed by 5 weeks with WD-CAC ± adenine and atherosclerotic lesions were quantified in the aortic valve. (**A**) Representative ORO staining of aortic root sections from control mice and (**B**) mice with adenine-induced nephropathy. (**C**) Quantification of aortic lesions in the aortic valve. NO represents the control group, fed with WD-CAC. Scale bar *=* 200μm. The data is plotted as a median with whiskers indicating the minimum and maximum values (*n =* 7–8). For statistical significance, ** *p* < 0.01.

**Figure 4 biomolecules-12-01147-f004:**
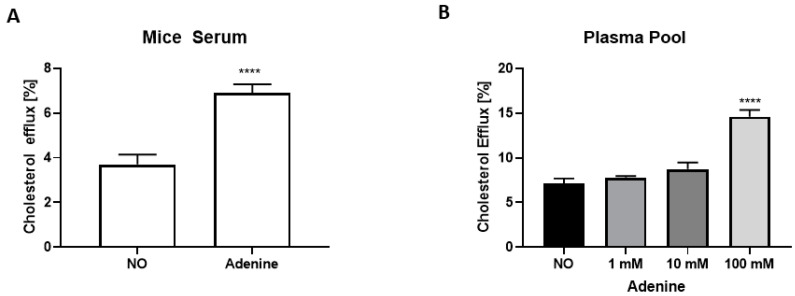
Adenine enhances cholesterol efflux. (**A**) Cholesterol efflux in RAW-365.7 cells using serum obtained from *ApoE* KO mice with normal renal function or with adenine-induced nephropathy. Data are means ± SEM (*n =* 9–11). (**B**) Cholesterol efflux in RAW264.7 cells pre-incubated with increasing concentrations of adenine (*n =* 4) with plasma pool as acceptor. NO represents the control group fed with WD-CAC. For statistical significance, **** *p* < 0.0001.

**Figure 5 biomolecules-12-01147-f005:**
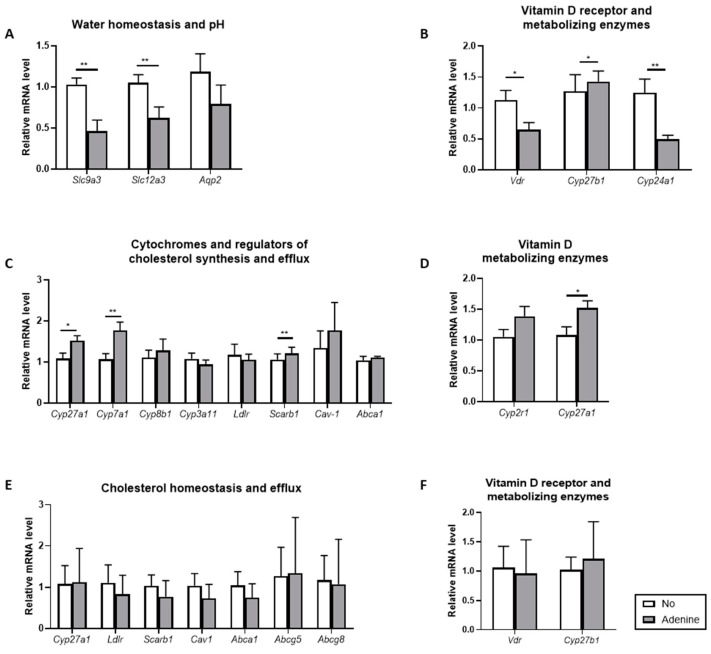
Adenine-induced effects on gene expression in the kidney and liver. (**A**) Expression of genes involved in Na^+^ transport and pH; (**B**) vitamin D receptor and vitamin D metabolizing enzymes were assessed in kidney tissues; (**C**) similarly, gene expression of genes accounting for cytochromes, regulators of cholesterol synthesis, and RCT, as well as (**D**) vitamin D metabolizing enzyme were investigated in the liver. (**E**) Expression of genes involved in cholesterol homeostasis and (**F**) vitamin D receptor and vitamin D metabolizing enzymes in the colon. Β-actin was used as a housekeeping gene. NO represents the control group fed with WD-CAC. Results are presented as means ± SEM (*n =* 8–11). For statistical difference, * *p* < 0.05, ** *p* < 0.01.

**Figure 6 biomolecules-12-01147-f006:**
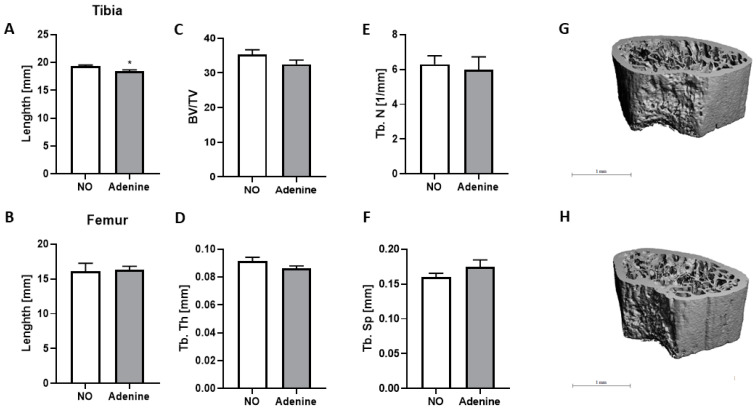
Bone analyses in 12 weeks old *ApoE* KO mice with normal renal function or adenine-induced nephropathy. (**A**) Length of tibias and (**B**) femurs were measured with a caliper. The distal part of the femurs was analyzed by μCT scan and computer-aided analysis: (**C**) bone volume (BV/TV), (**D**) trabecular thickness (Tb. Th), (**E**) trabecular number (Tb. N), and (**F**) trabecular space (Tb, Sp) are shown. Representative 3D lCT images of the femur from a mouse (**G**) with a normal kidney and (**H**) adenine-induced nephropathy. Scale bar *=* 1mm. NO represents the control group fed with WD-CAC. Data are shown as mean ± SEM (*n =* 8–9). For statistical significance, * *p* < 0.05.

**Figure 7 biomolecules-12-01147-f007:**
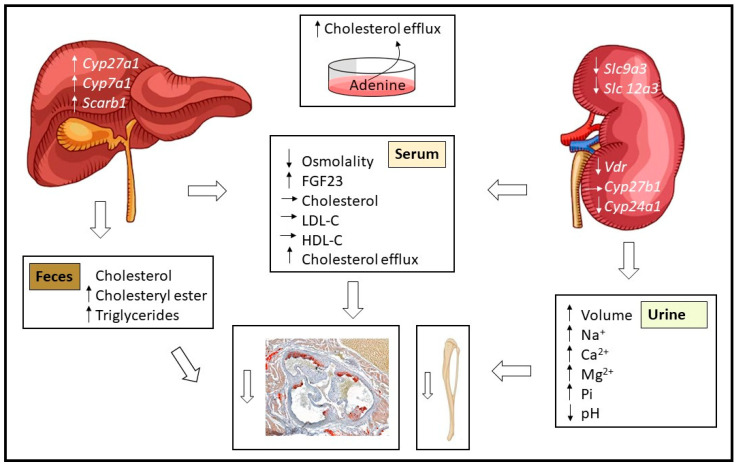
Possible mechanisms for the anti-atherosclerotic effect of adenine and consequences on bones in *ApoE* KO mice. Adenine induced tubular damage illustrated by increased FGF23 in serum and leading to changes in urinary excretion and composition. The kidney compensates by downregulating the transporters involved in Na^+^ transport. The expression of the genes accounting for vitamin D metabolism is also downregulated. Due to urinary Ca^2+^ and Pi loss, the growth of the tibia is slowed. Mice fed with adenine develop fewer atherosclerotic lesions. This phenotype is mainly driven by a serum more efficient to efflux cholesterol and by an increased fecal output. In the liver, genes involved in cholesterol efflux and homeostasis are upregulated. Adenine itself enhances cholesterol efflux in vitro.

**Table 1 biomolecules-12-01147-t001:** Effect of Adenine on marker of CKD in urine and serum in *ApoE* KO mice.

Parameter	NO	Adenine	*p* Value
	*n =* 10	*n =* 9	
URINE			
Creatinine [µmol/24 h]	3.33 ± 0.38	3.79 ± 0.27	NS
Na^+^ [µmol/24 h]	112.4 ± 13.9	175.4 ± 18.3	0.0125
K^+^ [µmol/24 h]	247.8 ± 34.05	308.4 ± 27.7	NS
Ca^2+^ [µmol/24 h]	1.01 ± 0.11	6.37 ± 2.07	0.0051
Mg^2+^ [µmol/24 h]	6.49 ± 0.71	12.54 ± 1.93	0.0032
Pi [µmol/24 h]	86.09 ± 11.87	102.1 ± 6.15	0.0236
Cl^−^ [µmol/24 h]	132 ± 18.33	163.8 ± 17.8	NS
Urea [µmol/24 h]	1271 ± 150.5	1432 ± 105.2	NS
Glucose [µmol/24 h]	1.51 ± 0.155	1.98 ± 0.24	NS
SERUM			
Creatinine [µmol/L]	36.6 ± 7.5	32.1 ± 9.0	NS
BUN [mg/dL]	39.28 ± 3.88	43.22 ± 5.32	NS
Osmolality [mOsmol/kg]	3492.0 ± 497.2	1026.0 ± 110.0	0.0003
FGF23 [pg/mL]	253.0 ± 92.3	1038.0 ± 403.5	0.0003
Glucose [mmol/L]	5.5 ± 0.26	6.6 ± 0.4	0.0272
RENAL FUNCTION			
Urea_Urine_/Urea_Serum_	156.3 ± 23.4	42.3 ± 8.1	0.0008
Creatinine_Urine_/Creatinine_Serum_	338.6 ± 201.8	112.4 ± 47.3	0.0011

Results are presented as means ± SEM with *p* value. NO represents the control group fed with WD-CAC and normal renal function. NS, not significant.

**Table 2 biomolecules-12-01147-t002:** Effect of adenine on lipid composition in serum, liver, and feces.

	NO	Adenine	*p* Value
	*n =* 10	*n =* 9	
SERUM			
Total Cholesterol [mg/dL]	503.3 ± 41.6	619.1 ± 51.3	NS
HDL [mg/mL]	0.08 ± 0.01	0.08 ± 0.01	NS
LDL [mg/mL]	2.24 ± 0.14	2.63 ± 0.21	NS
Triglycerides [mg/dL]	74.58 ± 15.94	93.73 ± 15.09	NS
HDL/LDL	3.54 ± 0.62	3.29 ± 0.51	NS
LIVER			
Cholesterol [µg/mg]	10.84 ± 0.73	9.74 ± 0.51	NS
Cholesteryl ester [µg/mg]	5.24 ± 0.81	4.79 ± 0.67	NS
Trigylcerides [nmol/mg]	14.64 ± 1.08	12.62 ± 1.27	NS
FECES			
Cholesterol [mg/24 h]	3.92 ± 0.52	5.21 ± 0.8	NS
Cholesteryl ester [mg/24 h]	0.35 ± 0.05	0.67 ± 0.12	0.0197
Trigylcerides [mg/24 h]	0.035 ± 0.007	0.149 ± 0.033	0.0042

Results are presented as means ± SEM with *p* value. NO represents the control group fed with WD-CAC and normal renal function. NS, not significant.

## Data Availability

All the data that support the findings of this study are available in the methods and results of this article.

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
