# Peer review of "Adenine-Induced Nephropathy Reduces Atherosclerosis in ApoE Knockout Mice"

_biomolecules, 2022, doi:10.3390/biom12081147_

Round 1

Reviewer 1 Report

General: The manuscript describes the axis between kidney function and vascular function, by using adenine food supplementation and ApoE mice. It is a limited study, straight forward with a surprising result. The experiments are clearly described and complete and the manuscript is well-written. There are a few points that need additional work/information.

Specifics: 

-       In the introduction, line 40, it is not clear whether this is rupture of the plaque or of the vessel.

-       In the description of the build-up of the atherosclerotic plaque I miss the role of endothelium. 

-       Primers (lines 158-162) are given by number without a reference where to find them.

-       Creatine concentration in serum and urine (table 1) are the same for controls and adenine-treated mice. Urea concentration in urine is not provided. The ratio, however, differs significantly. No explanation is given, neither in the result section nor in the discussion.

-       The size of the plaque is smaller in adenine-treated mice. Information on the cholesterol content would be valuable.

-       Apparently, more cholesterol is removed (from the blood) to the intestine of adenine-treated mice (Table 2). Enzyme expression values are given for enzymes of kidney and liver (fig 5c). What about the enzymes of the intestine?

All in all, an interesting paper with an unexpected result. M&M and result sections are largely adequate. Discussion and introduction are rather long but not excessive. 

Author Response

We thank the reviewer fof his constructive comments. Enclose please find a point by point answer to the reviewer in the file below:

Reviewer 2 Report

This is an interesting paper that aims to investigate whether kidney disease has an impact on the development of atherosclerosis. The authors have used adenine to induce renal tubular injury and they report that mice fed adenine had less atherosclerotic lesions. They then used in vitro studies to show that adenine enhances cholesterol efflux by cells.

This is an interesting study but there are some limitations that should be addressed in the manuscript.

Given the authors’ findings that adenine increases cholesterol efflux by cells, adenine-induced renal failure is not an ideal model to assess the interaction between kidney disease and atherosclerosis. This should be addressed in the discussion.

In Line 77 in the introduction the authors state: “In mice, there are two main strategies to induce CKD.” This is followed by the description of the 5/6 nephrectomy model and the adenine model that was used in this study. This wording gives the false impression that these are the only models of CKD, which is not accurate as a range of models have been developed (s

It is unclear why the authors chose to investigate bone phenotype in this study. The link between kidney disease and bone health is known, but the rationale to investigate this in the context of atherosclerosis is not clear.

It would be more appropriate to calculate the rate of creatinine clearance (ml/min per gram of body weight) rather than show the ratio of urinary creatinine to serum creatinine as an indicator of renal function.

Quantification of the morphological changes observed in the kidneys are needed to demonstrate the frequency of the features described and shown in the representative pictures in Figure 2.

It is unclear why a macrophage cell line (RAW264.7) was used to investigate cholesterol efflux.

Some grammatical errors and typos are noted:

Line 38-9: “This causes damages to cellular components” should be “This causes damage to cellular components”

Line 52: “they are wildly applied for the treatment of atherosclerosis” Do the authors mean widely applied?

Line 85-86: “These non-soluble materials are crystallized in renal tubules and cause renal damages” should be “These non-soluble materials are crystallized in renal tubules and cause renal damage”

Line 196: “One-way ANOWA” should be: “One-way ANOVA”

Author Response

We thank the reviewer fof his constructive comments.

Please find the answers to the reviewer in the following document:
